# Healthcare-seeking behaviors and factors influencing non-adherence among cervical cancer patients attending Bugando Oncology Clinic in Mwanza, Tanzania: A qualitative Phenomenological study

**Bashari Nuru Kidaya[1], Joseph Rogathe Mwanga[2]\*, Oscar Ottoman Muhini[3], Halima Mdemu Mwaisungu[4], Elisha Mabula Juma[5], Vivian Elikana Buremo[1]**

**1** Department of Oncology, Bugando Medical Centre, Mwanza, Tanzania **2** Department of Epidemiology, Biostatistics & Behavioral Sciences, School of Public Health, Catholic University of Health, and Allied Sciences (CUHAS) Mwanza, Tanzania **3** Department of Pathology, Weill Bugando School of Medicine, Catholic University of Health and Allied Sciences (CUHAS) Mwanza, Tanzania **4** Department of Medical Ethics, Weill Bugando School of Medicine, Catholic University of Health, and Allied Sciences (CUHAS) Mwanza, Tanzania **5** National Institute for Medical Research (NIMR), Mwanza Center, Tanzania

\* jrmwanga@yahoo.co.uk

## Abstract

### Background

In low- and middle-income countries, particularly in Tanzania, most patients with cervical cancer present with advanced-stage disease and exhibit non-adherence which results in increased numbers of patients with cancer-related deaths. The current study explores the health-seeking behavior of cervical cancer patients and the factors that influence their non-adherence to cancer care.

### Objective

To explore the healthcare-seeking behaviors and identify factors influencing non-adherence among cervical cancer patients attending Bugando Medical Center's Oncology Clinic in Mwanza, Tanzania.

### Methods

A qualitative phenomenological design was adopted to explore the lived experiences of 15 households with non-adherent cervical cancer patients, after obtaining patients' information from the chemo radiation treatment registries of Bugando Oncology Clinic in Mwanza, Tanzania. Data were generated through in-depth interviews with patients, and NVivo 12 qualitative computer software was used to aid analysis. Thematic content analysis was conducted to uncover underlying meanings and patterns in the data, providing valuable insights into the phenomena under investigation.

**Data availability statement:** All relevant
data are within the article and its Supporting
Information files.

**Funding:** The author(s) received no specific
funding for this work.

**Competing interests:** The authors have
declared that no competing interests exist.

### Findings

The study revealed poor adherence to treatment-seeking by most of the study informants
in the form of delayed health-seeking at nearby health facilities. The limited capacity for
correct cervical cancer diagnosis also pushed some of the informants to turn to self-
medication including traditional remedies. The findings revealed delayed healthcare-
seeking behavior and poor adherence to most of the study participants seeking medical
care at nearby health facilities with limited capacity for correct cervical cancer diagnosis.
Some turned to self-medications including traditional remedies. Financial constraints
emerged as a major obstacle, affecting the affordability of treatments, transportation, and
accommodation. Moreover, the malfunctioning radiotherapy machine posed a significant
barrier to effective treatment. Limited comprehensive information on their condition, treat-
ment options, and schedules further hindered adherence.

### Conclusion

There is a need for improved access to an appropriate healthcare system and inter-
ventions to improve non-adherence to treatment of cervical cancer services, as well as
increase public awareness, initiation of tools for good adherence of chemotherapy treat-
ment and mass screening of cervical cancer risk factors and earlier diagnosis for better
survival from the disease in Tanzania.

## Introduction

### Problem formulation

According to estimates, there were 604,000 new cases and 342,000 deaths from cervical cancer
in women worldwide in 2020[1]. Around 90% of new cases and fatalities globally in 2020 took
place in low- and middle-income countries (LMICs)[1]. Tanzania is no exception as cervical
cancer is the leading cancer in both sexes combined in terms of both incidence and mortal-
ity[2]. Among the estimated 40,464 individuals diagnosed with cancer in 2020, cervical cancer
constituted 25.3% of all newly identified cases[3]. In addition, out of the estimated 26,945
cancer-related deaths, cervical cancer accounted for 24.2% of all fatalities[3]. Bugando Medical
Center (BMC) Oncology Clinic was established in 2009 due to the increased number of cancer
patients being reported in the lake zone, it has served over 73,000 patients up to December
2020, with the majority being cervical cancer patients (14.1%)[4].

Correct cancer diagnosis and timely completion of the recommended course of therapy
are crucial in cancer management to ensure the desired outcome[5]. In Tanzania, cervical
cancer accounted for 39% of all newly developed cancers in women in 2018, and among them,
approximately 80% of cancer patients sought treatment when their condition was already
advanced, necessitating a combination of radiotherapy and chemotherapy for disease cure
or palliation[6,7]. The radiotherapy treatment plan entails five consecutive weeks of external
beam radiotherapy, administered daily from Monday to Friday, followed by brachytherapy
performed once every three consecutive weeks[8]. Concurrently, chemotherapy is adminis-
tered with external beam radiotherapy, comprising five cycles over five consecutive weeks,
with one cycle per week[8]. For overall survival as well as local tumor control, definitive
chemoradiation for cervical cancer must be completed within eight weeks[9].

Patients' adherence to these multitherapy schedules is influenced by patient-related factors,
therapy-related factors, condition-related factors, health system factors, and socioeconomic

factors[10]. From the patient's perspective, information and knowledge of diseases and their treatment, communication, trust in patient-provider relationships, support, and adequate resources proved to be the most important facilitators of medication adherence[10]. Enhanced adherence to cancer treatment brings benefits like improved treatment effectiveness, better disease control, increased survival rates, reduced recurrence risk, and improved quality of life[11]. Conversely, non-adherence reduces treatment efficacy, impacting quality of life and health economics, making it a significant public health concern[11,12].

While healthcare-seeking behaviors and non-adherence to cancer treatment are known challenges, the existing literature has not provided an adequate understanding of the healthcare-seeking behaviors and specific determinants impacting non-adherence among cervical cancer patients in the context of Tanzania's unique socio-cultural and economic dynamics. Therefore, this study intended to explore healthcare-seeking behaviors and factors influencing non-adherence among cervical cancer patients attending BMC Oncology Clinic in Mwanza, Tanzania.

Adherence to long-term therapy for chronic illnesses in developed countries averages 50%, while in developing countries, the rates are even lower[10]. Cultural beliefs and societal norms, health literacy, and access to healthcare services can influence the reluctance of individuals to seek and adhere to medical advice[13]. Non-adherence may lead to an incomplete response to therapy, an increased risk of disease progression, and decreased chances of successful treatment outcomes[14]. Consequently, this can strain the provision of cancer care services by wasting valuable healthcare resources and increasing the demand for advanced and costly treatments[15].

Efforts to impart public education, increase cancer care availability, and lower cancer care costs have not fully addressed healthcare-seeking behavior and medication adherence challenges. Only 10.8% of referred women seek cervical cancer treatment, and according to BMC's radiotherapy registries, 52.3% of cervical cancer patients who began radiotherapy from January to June 2023 did not complete their treatments within the required eight weeks[16]. Limited insight exists on Tanzanian cervical cancer patients' healthcare-seeking behaviors and factors influencing non-adherence. This study aimed to explore these behaviors, identify non-adherence factors, and suggest improvements for better patient outcomes in the region.

## Materials and methods

### Qualitative approach and research paradigm

This was a community-based qualitative study that employed an interpretive paradigm and phenomenological design. This design aimed to understand the lived experiences of participants (what all participants have in common as they experience a phenomenon of interest), which in this case was the healthcare-seeking behaviors and factors influencing non-adherence to clinical advice among cervical cancer patients attending the BMC Oncology Clinic.

The rationale for choosing the said design lies in the basic purpose of phenomenology, which is to reduce individual experiences with a phenomenon to a composite description of the essence of experience for all individuals. Thus, the study was able to explore deeply the experiences, emotions, and beliefs of non-adherent cervical cancer patients, enabling the researchers to gain insights into their healthcare-seeking behaviors and reasons behind their non-adherence to medical advice.

In this research, we took into consideration the process known as 'bracketing'[17]. Both types of bracketing, descriptive and reflexive were used in this study. Regarding descriptive bracketing that is we tried our level best to put aside our feelings, beliefs, and ideas about the phenomenon under consideration (to avoid biasing their observations), to see the experience

from the eyes of the cancer patients who have had lived the experiences. The goal of descriptive bracketing was to capture the essence of informants' experience as purely as possible. By doing so, we were able to gain deeper insights into lived experiences of individuals and understand the meaning they attach to those experiences. We also applied reflexive bracketing in the sense that we acknowledged that complete objectivity in research is impossible. However, we tried to eliminate biases by actively reflecting on our own influence on the research process. We were aware of how our backgrounds, experiences and assumptions might shape our interpretations and interactions with data. We were transparent and reflexive to make research process more rigorous by explicitly acknowledging and accounting for our roles as researchers.

As phenomenological studies examine human experiences through the descriptions of meaning provided by the people living the experience (in this case cervical cancer patients), the limitation implicit in this design is that bracketing personal experiences may be difficult for the researcher to implement. Thus, the researcher needs to decide how best and in what way his or her understanding will be introduced into the study.

The choice of phenomenology design in this study was expected to influence study conclusions and transferability. The transferability was achieved by giving a 'thick description,' that is detailed account of field experiences in which we made explicit the pattern of cultural and social relationships and put them in context[18]. The study was conducted from January to September 2023. The recruitment period started on 24/08/2023 and ended on 12/09/2023. This research is reported in accordance with the Standards for Reporting Qualitative Research (SRQR)[19].

## Researchers' characteristics and reflexivity

The research team which comprised the PI, (master's in public health student), the academic supervisors; (a social scientist with a doctorate), and (pathologist with a master's degree), and research assistants (with basic degree) had no prior relationship with the informants before the study commenced. More importantly, informants were informed of the purpose of the study through their local government leaders to foster recognition and trust. The characteristics that were reported about the research team were the reasons and interest in the research topic.

## Study context

The study was conducted in the households of families with a non-adherent cervical cancer patient in five regions based on data obtained from the chemoradiation treatment registries at BMC Oncology Clinic, BMC is situated in Mwanza City which is located on the southern shores of Lake Victoria in the north-western part of the United Republic of Tanzania. BMC serves as a zonal referral, consultant, and university teaching hospital for over 18 million people in the Lake and Western zones of Tanzania offering specialized care for eight regions including Mwanza, Mara, Kagera, Shinyanga, Geita, Simiyu, Tabora, and Kigoma.

## Sampling strategy

The sampling strategy was purposeful in the sense that we selected individuals and sites for study because we thought that they could purposefully inform an understanding of our research problem and central phenomenon in the study. Sampling was based on selecting individuals with experience of the phenomenon being studied. BMC Oncology Clinic registers were used to locate the study informants. The study involved 15 non-adherent cervical cancer patients from the following five regions namely Mara (4), Mwanza (6), Simiyu (1), Shinyanga

(2), and Tabora (2). At the 15th interview, no further sampling and data collection were necessary as saturation was reached[20].

### Ethical considerations

Ethical approval to conduct the study was obtained from the joint CUHAS/BMC research ethics and review committee. with Research Clearance certificate No. CREC/708/2023 was granted. Permission to conduct the study was obtained from relevant authorities at the regions, districts, and local levels. Written informed consent was obtained from all study informants after full disclosure through the information sheet of the informed consent document (S1 File). Informants were assured of privacy, anonymity, and confidentiality regarding the information collected for the study. Furthermore, all informants gave consent for interview transcripts to be published, and those transcripts did not contain any potentially identifying information.

### Data collection methods

In-depth interviews were held in the households of non-adherent cervical cancer patients, using an interview guide in *Kiswahili* featuring main and probe questions. The guide was designed and pre-tested by a research team prior to data collection. The guide had three sections: Participant's socio-demographic information (Section A), Exploring healthcare-seeking behaviors (Section B), and Identifying factors affecting non-adherence (Section C). Each interview lasted between 30 to 45 minutes. Interviews were conducted in *Kiswahili*, a common language spoken in Tanzania. Conduction of interviews at informants' households allowed us to contextualize data. Toward the end of the interviews, informants were invited to provide additional information and were allowed to ask questions, if any. A digital recorder was used for recording interviews with permission from informants. Research assistants also took notes to complement data documentation.

### Data processing and analysis

**Data organization.** Management and analysis of qualitative data was an iterative process[21]. All 15 audio files containing interviews were transferred into the computer from digital recorders, backed up in another computer in the field, and transcribed verbatim. *Kiswahili* transcripts were then translated into English for further processing (S2 File). The English transcripts were subsequently checked against the audio for the accuracy of the translation. Back translation was done by randomly selecting a few English transcripts and then translating them back to *Kiswahili*. Field notes taken during the interviews were subsequently typed up and were reserved for future use during data interpretation.

**Initial coding and coding strategy.** Transcripts were read repeatedly and coded by two investigators to maximize inter-coder variability using NVivo 12 qualitative data analysis software (QSR International Pty Ltd. Sydney, Australia). A hybrid coding was adopted, combining both deductive and inductive strategies[22]. Data was organized in a systematic and structured way by using topics in the interview guide to initially categorize and code the data (deductive coding); those topics became the initial codes. Then data was systematically and iteratively reviewed by looking at specific information related to topics to ensure an exhaustive set of data support each code. Data was also coded inductively[23], to allow coded to derive from the data and during this process, additional themes that were not captured by the interview guide were identified, and the internal validity and robustness of existing codes were confirmed (S3 File).

**Data reduction.** Data was then reduced into manageable sizes by identifying common themes, patterns, or categories. This involved grouping coded data into larger categories or themes. Data was summarized and condensed to highlight the most relevant and significant pieces of information. Thematic Content analysis was performed by two qualitative researchers.

**Data display.** Data was displayed in matrices. Both thematic and conceptual matrices were used to display data in a meaningful and accessible way to others. Quotations provided in this paper are illustrative of experiences reported by the informants and are used sparingly to support key themes.

**Conclusion drawing.** Conclusions were drawn from the study findings by two investigators interpreting the findings in the light of research question and objectives. This ensured reflection on their own assumptions and potential biases. This reflexivity enhanced credibility and depth of analysis. Contextual insights from the fieldnotes were key to interpreting the findings.

This study applied techniques to enhance the trustworthiness and credibility of data analysis by member checking in coding, data interpretation, and investigators' triangulation. Since the processing and analysis of qualitative data was systematic, explicit, and reproducible, the validation and trustworthiness of the findings were established[24].

## Results

### Socio-demographic characteristics of the study informants

We examined healthcare-seeking behaviors and factors affecting non-adherence among cervical cancer patients at BMC Oncology Clinic in Mwanza, Tanzania. These individuals included women aged 35–39 (n = 2, 13.3%), 40–49 (n = 3, 20.0%), 50–59 (n = 8,53.3%), and 60–65 (n = 2, 13.3%). Their education levels ranged from no formal education (n = 5, 33.3%) to primary school dropout (n = 3, 20.0%), primary school completion (n = 6, 40.0%), and secondary school dropout (n = 1, 6.7%). Marital statuses included single (n = 1, 6.7%), married (n = 6, 40.0%), separated (n = 1, 6.7%), divorced (n = 3, 20.0%), and widowed (n = 4, 26.7%). Religious affiliations varied, with informants identifying as having no religion (n = 2, 13.3%), Catholic (n = 6, 40.0%), Mennonite (n = 1, 6.7%), Pentecostal (n = 3, 20.0%), Baptist (n = 1, 6.7%), and Muslim (n = 2, 13.3%).

### Themes that emerged from the study

Findings from interviews revealed five main themes and fifteen subthemes emphasizing the need for support for existing systems to enhance access to cancer care services and provide cancer education for cervical cancer patients attending BMC Oncology Clinic in Mwanza, Tanzania. In this paper, the bolded and italicized sub-headings refer to the main themes and the italicized ones (apart from those in quotations) refer to the sub-themes (S4 File).

**Cancer Awareness among informants.** This theme provides insights into the general awareness and knowledge base of individuals regarding cancer-related information. The theme encompasses various aspects such as the types of cancer they are aware of, their understanding of specific cancers like cervical cancer, the associated symptoms, risk factors, and available treatments. It also includes their awareness of preventive measures, screening methods, and the importance of early detection.

**Awareness of cancer types.** The informants demonstrated varying levels of familiarity with different types of cancer. Some were more knowledgeable about specific types, while others had a more limited understanding of various cancers. This subtheme highlights the diversity in awareness levels among the informants regarding different cancer types. One informant remarked:

*"There are various types of cancer including cervical cancer, prostate cancer, throat cancer, lung cancer, leukemia, brain cancer, skin cancer, bladder cancer, colon cancer, and anorectal cancer"* (Cacx Non-Adherence 06).

Some informants stated that they became aware of the existence of certain types of cancer when they sought treatment at BMC:

*"I am familiar with cervical cancer, and I came across others like neck tumors, throat cancers, and eye cancers during my time at Bugando for treatments"* (Cacx Non-Adherence 09).

On the other hand, another informant confessed that she didn't know any type of cancer

*"I don't know"* (Cacx Non-Adherence 05)

**Awareness of cervical cancer symptoms.** Informants demonstrated varying levels of awareness regarding the symptoms associated with cervical cancer as they responded, referring to their personal experiences with the disease. Some could identify specific symptoms:

*"I had severe bleeding and a tumor, ulcers in private parts, vaginal discharge"* (Cacx Non-Adherence 08).

Another informant described that

*"I had postcoital bleeding, smelling discharge, waist and abdominal pain, my private parts were itching and burning sensation"* (Cacx Non-Adherence 02).

Other informants provided more general descriptions as follows:

*"Non-stop bleeding"* (Cacx Non-Adherence 15).

This suggests a range in the depth of knowledge about cervical cancer symptoms among the informants.

## Healthcare-seeking behaviors

In this theme informants reported different experiences in seeking relief for their conditions with the majority going directly to the hospital, while others tried traditional remedies and self-drugs. Stigma was mentioned as among the social driving factors to seek healthcare as pointed out by one informant:

*"I had severe pain, bleeding, and a bad smell. As a woman, it is uncomfortable to have smelling discharge. It was difficult for me to sit in the church as people could sense my smell, and when I stood up my clothes looked wet with blood. When church fellows come to my home, I told them I am smelling badly"* (Cacx Non-Adherence 02).

Some informants said that family members were the reason behind their decision-making process to seek hospital cancer care. Family members also played an important role in providing emotional support to help alleviate the psychological stress that the informants were experiencing after receiving a cervical cancer diagnosis. The following are the informants' remarks:

*"Just have some medicine. I didn't have faith in going to traditional healers because even my grandmother had cervical cancer. I just went to my auntie who cared about my grandmother and told her about my condition. She told me that my symptoms resemble those of my grandmother, she encouraged me to go to the hospital, and that my grandmother underwent a hysterectomy, and her life went on well, and died from another illness. My auntie encouraged and insisted on me, then I stopped crying, and then I went to the hospital"* (Cacx Non-Adherence 03).

*"The family decided that I must be taken to the hospital. Therefore, we went to the district hospital, where they told my son that they suspected me to have cervical cancer and referred me to Bugando"* (Cacx Non-Adherence 08).

One informant decided herself, not to wait for symptoms to appear, she just went to a screening camp and was suspected to have cervical cancer. The following were her remarks:

*"There was a screening camp in my village, I went there where they screened me, and they told me that I have the indications for a tumor. I was told to go to the hospital, but I didn't have money, so I didn't go. After a month, I had severe bleeding, I went back to the hospital, and they told me I was suspicious of cervical cancer. They asked me to pay Tsh. 100,000 which is roughly equivalent to 40 USD for the biopsy test. I didn't have the money and told them to let me find it. After a month I went back, and they told me to go to Malolo Hospital, where they told me that I have cervical cancer"* (Cacx Non-Adherence 05).

A few informants explained that they started with traditional medicine, but they didn't get any relief, thereafter they went to the hospital:

*"I was given the traditional medicine locally called "nyanjugo", which is administered by drinking and by sitting in the mixture of that medicine and water"* (Cacx Non-Adherence 02).

*"During the period of one month when I was waiting for my final diagnosis at Bugando, I shared this information with my friends, who provided me with many traditional medicines, but I didn't get any relief"* (Cacx Non-Adherence 07).

*"As the traditional medicine didn't work that's why we went to the hospital"* (Cacx Non-Adherence 12).

## Challenges faced in seeking healthcare for cervical cancer

Many patients stated that misdiagnosis and resulting mistreatment was a common challenge they faced. Their nearby hospitals had limited capacity to diagnose cervical cancer, leading to the misinterpretation of their symptoms as bacterial infections. As a result, some received treatment only for the symptoms they presented:

*"My early visit after the onset of symptoms at the hospital I was told that I have UTI and typhoid and given the respective medicines, but I got no relief"* (Cacx Non-Adherence 07).

*"I was being treated as I am having UTI, typhoid or fungus"* (Cacx Non-Adherence 09).

*"I never tried to treat myself, I went straight to the hospital in Kenya, and they injected me a syringe to stop bleeding and gave me some drugs, and the bleeding stopped. After a year I felt stomach pain and heavy bleeding, I went back to the same hospital and they told me it was cancer, and then they referred me to a hospital in Musoma"* (Cacx Non-Adherence 01).

*"I just went to a private hospital where I was sometimes told that I have pelvic inflammatory disease, and sometimes I was told I had urinary tract infection (UTI) that has reached 60. When I had heavy bleeding, and sometimes severe anemia that made me fall, I told my daughter that let's go to Butimba, the government hospital, as I don't get any better in the private hospitals"* (Cacx Non-Adherence 03).

Financial constraints appeared to be a challenge faced by many informants. Since many of them reside hundreds of kilometers away from BMC, being referred there exacerbated their already significant burden of dealing with cervical cancer:

*"I had no money to go to Bugando as we always heard that to go to Bugando someone must have money"* (Cacx Non-Adherence 02).

*"I had no money Tsh. 80,000 (roughly equivalent to 32 USD) for cervical cancer diagnosis at Bugando so they took a sample and told me to find the money to get my diagnosis. I went back home and felt more severe bleeding thus I went to Buzuruga Health Center, they admitted me and told me that I had bacteria in my uterus. They told me to buy medicine for Tsh. 80,000 (roughly equivalent to 32 USD) but I only afforded half a dose. The situation kept worsening and I was referred to Sekou Toure Regional Referral Hospital, where the money was also a challenge, but finally, we managed to get the money and diagnosed with cervical cancer. My husband went for my diagnosis report to Bugando and when we showed it to Sekou Toure they referred me to Bugando for cervical cancer treatment. So, my biggest challenge was financial constraints"* (Cacx Non-Adherence 10).

*"I went to a dispensary but did not know what I was suffering from, then I went to a hospital in Kenya, where they told me to be referred to another hospital in Kenya, and they gave me a painkiller drug. I told them that the hospital was very far, so I decided to go to Shirati Hospital after a week. At the Shirati hospital, a nurse screened me and said that the disease looked like cervical cancer, so she had to refer me to BMC. I told them I didn't know anyone at Bugando, then who would help me and where I get the money as I have neither mother nor child to rely on. They told me to go and look for money and if I got enough money, I should return to have a referral letter. I told my church fellows, and they contributed some money towards my treatment, and I added up mine, thereafter I went to Shirati hospital where they gave me a referral letter to Bugando"* (Cacx Non-Adherence 02).

On the other hand, informants were confused about whether cancer services are provided for free or if they must pay for them. They received conflicting messages about cost-sharing for cancer care services from different healthcare providers. One informant said:

*My treatment is going well but when we got to Bugando, the doctor told us to find money, Tsh. 328,000/ = (roughly equivalent to 131 USD). We then went back home as the machine was broken down. We got back to Bugando after the machine was fixed. Then another doctor told us that treatment for cervical cancer is free, then after a while, we were told that money is needed to consult a doctor Tsh 5,000, (roughly equivalent to 2 USD) when you are going for blood test Tsh. 17,000/ = roughly equivalent to 7 USD., to have chemotherapy money is needed, for brachytherapy money is needed more than Tsh. 300,000/ =. roughly equivalent to 120 USD). How is it for me as I am broke, sometimes I failed to get chemotherapy as I did not have money but the day after, they helped me by providing free chemotherapy to me"* (Cacx Non-Adherence 02).

The malfunction of the machine and it is reaching its full capacity also posed a challenge for some informants. They managed to arrive at BMC promptly after being referred but encountered difficulties in starting radiotherapy on time as follows:

*"I had nonstop bleeding from December 2022 to June 2023 when I had a hysterectomy at Bugando. I went to Nyamagana District Hospital, Butimba where they diagnosed me with cervical cancer, and they referred me to Bugando where I had some tests and underwent a hysterectomy on 02 May 2023. Then my doctor referred me to the Oncology Department, where the doctor read my history and told me about treatments in the Oncology Department. Unfortunately, I was told that the machine is broken down, so I will have to wait for a month to start radiotherapy treatment"* (Cacx Non-Adherence 03).

*"At Bugando, I was told to wait for two weeks to start radiotherapy as the machine has reached its maximum capacity"* (Cacx Non-Adherence 08).

### Overcoming the challenges

It was revealed that some informants had their family members by their side to support them financially, while others had to rely on their fellow church members to contribute a top-up so that they could attend the cervical cancer care services at Bugando Medical Centre.

*"My daughter supported me financially"* (Cacx Non-Adherence 04).

*"My church fellows encouraged and contributed some money towards my treatment, and I add up mine"* (Cacx Non-Adherence 02).

### Perceptions of cervical cancer and the care services

This theme deliberates on the beliefs, attitudes, and opinions that women held regarding the treatment of cervical cancer. This encompasses their understanding of available treatment options, their expectations, fears, and overall outlook toward the process of receiving medical care for cervical cancer. This theme explores the subjective experiences and viewpoints of women as they navigated through the treatment pathways, shedding light on their perspectives and providing valuable insights into how they perceived and acted upon their conditions.

### Advice to other cervical cancer symptomatic women

In this sub-theme informants stated clearly that they would advise women who have cervical cancer symptoms to go to the hospital for cancer care and use themselves as living testimonies. The following quotations are examples:

*"I will just advise her to come straight to the hospital, as you can see, I am now doing well to the extent of walking around without wearing underwear"* (Cacx Non-Adherence 03).

*"I will advise her that cervical cancer is treatable, she must go the hospital. I will testify using myself as an example of someone who has healed from cervical cancer"* (Cacx Non-Adherence 04).

*"I will advise her to go to Bugando"* (Cacx Non-Adherence 02).

Some informants had already started being cancer ambassadors to other patients who hesitate to go to the hospital as indicated by the following quotation:

*"I have started to tell some patients with cancer symptoms to just go to the hospital to check for cancer"* (Cacx Non-Adherence 06).

## Misconceptions about hospital cancer care

Informants shared the misconceptions that were circulating in their communities. They now realize that these misconceptions were incorrect and were misleading patients, causing them to avoid seeking cancer care at hospitals. One informant lamented that:

*"Cervical cancer is not treatable; Radiotherapy involves the use of fire to burn a body part"* (Cacx Non-Adherence 04).

Other informants were very bold enough to show their trust in hospital cancer care:

*"We always hear that the disease that cannot be treated at Shirati Hospital cannot even be treated at Bugando. Going to Bugando will lead to only death." I just said even if I die at Bugando, my church fellows will come to take me"* (Cacx Non-Adherence 02).

## Informant's initial thoughts when diagnosed with cervical cancer

Subtheme 3 revealed that informants had varying emotional experiences after receiving their cancer diagnosis. Some stated that they believed their days were numbered, as cancer is a disease that instills fear in everyone. They observed that the majority of those who contracted cancer ended up passing away shortly after being diagnosed. The following quotations are self-explanatory:

*"My heart was beating very fast, I didn't have peace of mind, as days' back we used to know that someone with cancer will die very soon, so I felt like my days are numbered"* (Cacx Non-Adherence 05).

*"I felt too much pain, I saw that my life had come to an end. You know, in my region this disease is not familiar, everyone I shared the information with that I have this disease, they felt afraid of me. This made me feel very nervous and I thought that I was going to die"* (Cacx Non-Adherence 01).

*"I didn't think anything rather than going for the treatment so that I can be healed"* (Cacx Non-Adherence 09).

Some informants found comfort within themselves especially after being told that the disease is treatable, while others received support from healthcare providers. In addition to concerns about the disease itself, they were also anxious about how they would manage the hospital bills and treatment-related expenses:

*"The nurse asked me are you ready to receive your diagnosis, are you alone? I told her that the one who accompanied me to Bugando is not my relative but just my neighbor, she is helping me as she works in Mwanza, so I am ready to get my diagnosis alone. The nurse told me that I had cervical cancer, I had some thinking, and she said sorry to me, I then felt some relief. I asked her if it was treatable, and she said yes, it was treatable. Then I felt some hope that the cervical cancer is treatable"* (Cacx Non-Adherence 02).

*"My daughter didn't tell me because she knows that I have low blood pressure. The doctor told me that I should not be afraid because I have pressure, I told him to just tell me as diseases are part of everyday life. After he told me I had cancer, I just wondered, how did I have it? I just believed that God would help me. I went to Bugando and saw children with eye cancer, I felt myself healed and I asked myself if children like these can have cancer, who am I not to have cancer? From that day I saw cancer as a normal disease"* (Cacx Non-Adherence 06).

*"I thought about how we will afford the treatments as my family is not well financially. But I thank God"* (Cacx Non-Adherence 10).

*"Firstly, I had a challenge with my marriage, I got divorced without any genuine reason. I kept it aside and prayed to God that I need my health as I have three children, one is in standard five, the second one is in form two, and the third one has completed university education yet to get employed. All are girls. Also, I was born alone, and my mother died, and my father has another family. Then who will help me with my treatments? So, I was very much stressed. My treatment looked difficult; my life was difficult. By then, I was playing the role of both a mother and a father and needed to hustle. I thank God that by the time my marriage started to be on the rock, I was doing business, and I lived within a very tight budget. I borrowed some money from the lending groups to build a house. Now that I am living in my house, I don't have to pay house rent but feeding my children, paying the school fees, and affording my treatments are challenges which I must cope with."* (Cacx Non-Adherence 03).

## Adherence to cancer treatments

This theme reflects varying levels of adherence among informants. Some struggled due to a limited awareness of their prescribed treatments while others had a clear understanding. Furthermore, adherence is influenced by awareness of treatment schedules, with some patients informed about their appointments and others not. Additionally, knowing specific conditions to follow during treatment plays a role, with some adhering while others may not, due to this awareness gap.

## Awareness of cancer treatments

The findings revealed varying levels of awareness among the informants regarding available treatment modalities. Some informants were familiar only with radiotherapy, while others were aware of chemotherapy. However, some individuals had limited awareness about any form of cancer treatment. This theme reflects diverse perspectives and levels of understanding among the informants regarding the available treatment options for cervical cancer. The following were the answers to the question of awareness of cancer treatment from some informants:

*"Chemotherapy and Radiotherapy"* (Cacx Non-Adherence 15).

*"I was told that I have cervical cancer so I should go for radiotherapy"* (Cacx Non-Adherence 09).

*"I don't know"* (Cacx Non-Adherence 1).

Some informants relied on fellow patients from the same tribe as their primary source of information regarding the treatment of their condition:

*"I was not informed by the healthcare provider. I met a woman from my tribe, who told me she had the same disease, and she is going on with the treatments. She told me that treatment is not a syringe, it is a machine. You just sleep on the bed and the machine delivers the treatment"* (Cacx Non-Adherence 02).

### Awareness of their treatment schedule

This second subtheme revealed that some informants were aware of their treatment schedules while others were not.

*"It was planned that I should have radiotherapy for 25 days and then brachytherapy"* (Cacx Non-Adherence 10).

*"External Beam Radiotherapy for 25 days followed by brachytherapy for 3 times in three weeks Cacx"* (Non-Adherence 14).

*"I was told that my radiotherapy will take 25 days, and I have to go for laboratory tests on Thursdays and get the results on Friday, to have chemotherapy on Mondays"* (Cacx Non-Adherence 09).

*"I don't know, I just get treatments"* (Cacx Non-Adherence 05)

### Challenges faced in adherence to treatment

In this third subtheme it was revealed that informants faced various challenges that hindered their full adherence to treatment recommendations and schedules. Some failed due to a misunderstanding of information, while others struggled due to the severity of disease symptoms and the side effects of cancer treatments: *"I was anemic, therefore stopped radiotherapy for one week"* (Cacx Non-Adherence 12).

*"I experienced some disturbances that required moving from one office to another due to unclear information about paying consultation fees"* (Cacx Non-Adherence 01).

*"My body was feeling not well, and I had no food appetite Cacx"* (Non-Adherence 09).

*"I experienced diarrhea, burning sensation as the side effects of chemotherapy"* (Cacx Non-Adherence 12).

### Conditions that informants were instructed to follow

In this fourth subtheme informants provided diverse explanations about the specific conditions that they were advised to follow during their cancer treatment regimen, reflecting a range of perspectives on managing their healthcare. Some of those were as follows:

*"To consider eating well, consuming fruits and vegetables, and drinking plenty of water"* (Cacx Non-Adherence 10).

*"We were told to eat well, a lot of fruits and plenty of water"* (Cacx Non-Adherence 07).

*"To consult a doctor on Thursdays, to do some laboratory tests before coming for chemotherapy on Mondays"* (Cacx Non-Adherence 08).

*"Frequently eating to maintain my blood level"* (Cacx Non-Adherence 14).

*"I was told not to be under the sun, not to be around fire. I lost appetite"* (Cacx Non-Adherence 09).

## Understanding the importance of adhering to treatment plans

The fifth subtheme revealed that informants shared a common understanding of the importance of adhering to cancer treatments. This adherence led to an improvement in their well-being, relief from symptoms, and, in some cases, complete healing from the disease.

*"Adhering to treatment is important to control the symptoms and get healed"* (Cacx Non-Adherence 09).

*"There is a benefit because I see some positive changes"* (Cacx Non-Adherence 15).

*"I adhere to my treatments; I am feeling better now though I feel some stomach pain sometimes. I shared with my doctor, and we are proceeding well"* (Cacx Non-Adherence 01).

*"I felt much better, no pain and the severe bleeding turned to be little, and no smelling"* (Cacx Non-Adherence 02).

*"Honestly speaking, I feel better, although I had some sores and fatigue because of radiation"* (Cacx Non-Adherence 03).

## Materials or teachings received to help improve their adherence

Subtheme six revealed that informants have had varying experiences regarding the provision of materials and teachings aimed at enhancing their adherence. The following quotations from some informants:

*"I once attended the teaching"* (Cacx Non-Adherence 15).

*"I attended the teachings, but I didn't get the flyers as only a few copies were given, and I didn't get one. Please if you have one, give me"* (Cacx Non-Adherence 09).

*"We have given some flyers but was very few, so I once read them. I always attend the teachings"* (Cacx Non-Adherence 05).

Some informants shared the challenges they faced in attending the teachings provided on Wednesday mornings at the radiotherapy unit and weekdays at the consultation clinic. One informant lamented:

*"For me, when I didn't have chemotherapy, I went to Bugando at 04:00 pm. Going early was a bit challenging as I stay away from Bugando, so usually I ended up getting number 60 out of 80. Waiting until my number was called for treatment was difficult for me as to where I eat, and how I survive the day while my life was this difficult. So, I attended the teachings only in the first week. After realizing that the teaching schedule is not friendly for me, I decided to come in the evening, so I can eat at home, and have enough rest before coming for radiotherapy"* (Cacx Non-Adherence 03).

## Need to know concerns

This theme suggests that the participants expressed a desire to be informed about specific aspects related to their condition or treatment. They raised questions or voiced uncertainties

about certain elements of their healthcare, indicating a strong need for clear and accessible information to address their concerns. This theme underscores the importance of effective information, and education communication in the context of cervical cancer care.

**Knowing about the causes of cancer.**  Informants were eager to understand the origins and causes of cancer, especially as the disease was not prevalent in earlier times. Some also appealed to the government to find a vaccine for the disease.

*"What causes cancer? Has the government failed to identify the root causes of cancer so that we can take measures to prevent it? We can observe that the disease is affecting an increasing number of people as time passes"* (Cacx Non-Adherence 9).

*"I wish the government could reduce the cost for the service, especially for the poor ones, as the diseases are affecting a lot of women. The government should find a vaccine to protect other women"* (Cacx Non-Adherence 10)

**Knowing the fate of their persisting symptoms.**  Some informants still experienced symptoms even after completing their planned treatments, although with less severity than before. They were eager to know why they still have the symptoms, when the symptoms will completely disappear, and when they will fully recover.

*"I want to know why I am still having vaginal discharge"* (Cacx Non-Adherence 08).

*"When will my fistula stop?"* (Cacx Non-Adherence 12).

*"I just want to ask, after I have completed all the treatments, will I be healed completely?"* (Cacx Non-Adherence 03).

## Discussion

### Introduction

We explored the healthcare-seeking behaviors and factors contributing to non-adherence among cervical cancer patients at BMC Oncology Clinic in Mwanza, Tanzania. Several comparable studies from various regions have produced consistent findings, illuminating the complex aspects of healthcare-seeking behaviors and the factors influencing non-adherence to cancer care services[25–27]. The outcomes of this study will contribute to the existing knowledge in this area.

### Cancer awareness among informants

The study revealed that informants had varying levels of awareness of different types of cancers including cervical cancer. However, the study revealed that most cervical cancer patients do not perform the screening as it is medically recommended that general population women should start human papillomavirus (HPV) testing at 30, repeating every 5 to 10 years. HIV-positive women begin at 25, with 3-to-5-year intervals[28].

Our findings depart from a previous qualitative study conducted in Tanzania, which aimed to assess Tanzanian women's knowledge about Cervical Cancer[29]. That study indicated that women who were least likely to be aware of cervical cancer were those in rural areas, with lower socioeconomic status, and low levels of education. In contrast, our study has shown that women are aware of the existence of cervical cancer and its associated symptoms. Awareness of cervical cancer appears to have increased among women because of ongoing cervical cancer awareness campaigns nationwide.

However, this awareness was not reflected in screening uptake which remains a challenge, as was also highlighted in the aforementioned study. Limited screening in Tanzania may lead to late-stage diagnoses, resulting in more advanced and potentially less treatable cases, thereby increasing mortality rates, burdening the healthcare system, and raising healthcare costs due to preventable cases and reduced productivity.

## Healthcare-seeking behaviors

It was evident that most women affected by cervical cancer demonstrated a preference for seeking cancer care services directly from hospitals, whereas only a small fraction considered traditional medicine or self-prescribed remedies. This shift in healthcare-seeking behavior reflects an evolving awareness of the critical role that professional medical care plays in the management of cervical cancer. Interestingly, our findings depart from those of a prior study conducted at the Ocean Road Cancer Institute, which indicated that a significant proportion of women experienced delays in accessing cancer care services due to engagement with traditional healers[30]. This discrepancy underscores the dynamic nature of healthcare preferences and decisions, which are influenced by a multitude of contextual factors.

A notable barrier in our study was the issue of misdiagnosis and mistreatment in nearby hospitals, which resulted in delays in receiving appropriate cancer care services. This can potentially worsen patient outcomes and increase the burden on the healthcare system, highlighting the urgent need for improved diagnostic accuracy and treatment protocols in the country. Addressing this challenge is crucial in ensuring that cervical cancer patients receive timely and accurate diagnoses, subsequently leading to more effective treatment outcomes.

A few informants who relied on traditional or self-medication experienced delays in receiving appropriate medical care, potentially resulting in more advanced and less treatable cases. Moreover, these remedies may not effectively manage cervical cancer and could lead to complications or mask symptoms, making accurate diagnosis and treatment more challenging. Early detection of cervical cancer through regular screening and proper medical care is crucial for effective treatment.

Depending on traditional or self-medication may lead to missed opportunities for early intervention. Additionally, delayed, or ineffective treatment can escalate healthcare costs, as more extensive and costly interventions may be necessary for advanced cases. This reliance on alternative treatments also adds strain to the healthcare system, as patients may only seek medical attention when their condition worsens, placing an additional burden on healthcare resources.

## Perception of cervical cancer and the care services

The negative perceptions harbored by the informants regarding cervical cancer and cancer in general before seeking care at BMC were largely shaped by prevailing misconceptions and the unfortunate experiences they had witnessed in their communities, where a significant number of cancer patients succumbed to the disease shortly after diagnosis. This unattractive reality induced considerable emotional and psychological distress among the informants, as they faced the fear of an imminent demise. This narrative resonates with the findings of a prior study conducted at Ocean Road Cancer Institute which similarly highlighted the prevalence of negative perceptions towards cancer treatments among participants.

Nevertheless, patients' perceptions underwent a notable transformation upon their arrival at BMC. They were exposed to a different reality, such as cancer education, recovery, and healing, including some of the informants themselves. Witnessing these positive outcomes contributed to a shift in their perspective towards a more optimistic outlook on cancer

treatment. The tangible evidence of patients overcoming the disease instilled a newfound sense of confidence and trust in the medical interventions provided at B M C. This shift in perceptions, from despair to hope, not only alleviated their initial fears but also empowered them to approach their treatment schedules with greater determination and positivity. This transformation underscores the crucial role that positive treatment outcomes and firsthand experiences play in reshaping individuals' perceptions of cancer care.

## Adherence to cancer treatments

Treatment adherence among cervical cancer patients at BMC Oncology Clinic in Mwanza, Tanzania, faced significant challenges, primarily attributed to financial constraints, machine malfunctions, severe cancer symptoms, and severe treatment-related side effects, notably anemia. These findings are like those of a study conducted in Uganda, which aimed to explore treatment adherence to concurrent chemo-radiation[31]. The study revealed that most informants missed treatments, citing factors such as disease stage and distance from the treatment facility.

Financial constraints emerged as a pervasive barrier to adherence. Many patients hailed from remote areas, hundreds of kilometers away from BMC. This geographical distance compounded the economic burden, as patients not only had to cover the costs of treatments but also transportation and accommodation. Many informants expressed apprehension about how they would manage to meet the costs associated with hospital bills and treatment expenses. Financial constraints can lead to delayed or inadequate treatment, potentially resulting in more advanced and less treatable cases. Additionally, it may lead to increased financial strain on families and communities, impacting their overall well-being and economic stability. Furthermore, it highlights the need for accessible and affordable healthcare services and support systems for cancer patients in the region. Addressing these financial constraints is crucial for improving the overall health outcomes and quality of life for cervical cancer patients in Tanzania.

The malfunctioning of the radiotherapy machine at BMC also presented a formidable obstacle to timely and effective cervical cancer treatments. For patients who managed to overcome financial barriers and make the journey, the frustration of encountering a non-functional machine was disheartening. This setback not only disrupted their treatment schedules but also extended the overall duration of their treatment journey, potentially impacting the effectiveness of radiotherapy for cancer patients. Additionally, it may result in increased stress and anxiety for patients, as well as additional logistical challenges in coordinating their care. The combination of financial strain and technical difficulties created a compounding effect, exacerbating the challenges of adherence. Furthermore, this issue underscores the importance of regular maintenance, backup equipment, and investments in healthcare infrastructure to ensure uninterrupted and effective cancer treatment services. Addressing the malfunctioning machine is crucial for maintaining the quality and continuity of care for patients at BMC.

Limited awareness regarding available treatments for cervical cancer, including which specific treatment is prescribed for them and the schedule for receiving it also posed a challenge to treatment adherence. This indicates a significant gap in information dissemination and patient education within the context of cervical cancer management. This may lead to delayed or ineffective treatment, potentially resulting in more advanced and less treatable cases. Additionally, this limited awareness may cause increased stress and anxiety for patients, as well as logistical challenges in coordinating their care. The informants' uncertainty highlights the urgent need for improved patient education and communication between healthcare providers and patients, ensuring they are well informed about treatment options and plans. Bridging

this knowledge gap is crucial for facilitating informed decision-making and promoting better adherence to treatment regimens among cervical cancer patients thus enhancing the overall quality of care and outcomes for cervical cancer patients in Tanzania.

Moreover, severe cancer symptoms and treatment-related side effects, particularly anemia, further complicated adherence efforts. Anemia, characterized by a deficiency of red blood cells, can lead to fatigue, weakness, and reduced capacity for physical activity[32]. For patients already struggling with the physical toll of cancer, anemia added an extra layer of burden, making it even more challenging to adhere to treatment schedules. The implications of severe cancer symptoms and treatment-related side effects, especially anemia, further complicating adherence efforts, have significant repercussions for cervical cancer management in Tanzania. They can lead to increased suffering and reduced quality of life for patients. Additionally, these challenges may necessitate additional healthcare resources and interventions, potentially straining the already fragile health system. Addressing severe symptoms and side effects becomes of paramount importance to improving the overall well-being and treatment outcomes of cervical cancer patients in the country. This underscores the importance of comprehensive care and support for those undergoing cancer treatments in Tanzania.

## Need to know concerns

Cervical cancer patients at BMC Oncology Clinic exhibited a strong desire for information regarding their condition and treatment outcomes. Many expressed a keen interest in understanding what to expect after completing their prescribed treatment regimens. They sought reassurance about the lingering symptoms and hoped to grasp a realistic timeline for their eventual subsidence. This yearning for clarity reflects a deep-seated need for a sense of control and predictability over their health journeys. Moreover, a significant number of patients voiced curiosity about the root causes of cervical cancer. They yearned for insights into the factors that may have contributed to their diagnosis. This curiosity was driven by a desire to comprehend the origins of their conditions, potentially as a means of preventing recurrence or safeguarding the health of loved ones. It also signaled a readiness to demystify a disease that often carries a shroud of fear and uncertainty.

The quest for knowledge by cervical cancer patients was not merely an unplanned inquiry, but a fundamental aspect of their coping mechanism. Being armed with information provided them with a semblance of empowerment and a foundation upon which they could construct a path toward recovery and wellness. It was evident that access to accurate, comprehensible, and timely information was fundamental in enabling these patients to navigate their cancer journeys with a greater sense of agency and resilience. This pervasive need for information highlights the critical role of patient education and counseling services in the comprehensive care of cervical cancer patients. Addressing these concerns can significantly enhance the patients' overall experience, diminish anxiety, and ultimately contribute to more informed decision-making regarding their treatment and aftercare.

## Policy implications

The findings that key barriers to cervical cancer care in Tanzania, included inadequate healthcare infrastructure, financial constraints, equipment malfunctions, and poor patient education underscore the need for health policy to focus on improving diagnostic capacity, ensuring financial support, maintaining treatment equipment, and enhancing patient education to improve treatment adherence and outcomes.

The study revealed key barriers to cervical cancer care in Tanzania, including inadequate healthcare infrastructure, financial constraints, equipment malfunctions, and poor patient

education. Health policy should focus on improving diagnostic capacity, ensuring financial support, maintaining treatment equipment, and enhancing patient education to improve treatment adherence and outcomes.

## Scientific contributions

The findings from this study expand the body of scientific knowledge by identifying specific, real-world barriers to cervical cancer care in Tanzania, such as limited diagnostic capacity, financial constraints, and equipment malfunctions. These insights deepen the understanding of how systemic factors contribute to treatment non-adherence. Additionally, the study highlights the crucial role of patient education in improving adherence, providing evidence that can inform targeted interventions. By focusing on the unique challenges in a low-resource setting, the research enriches global discussions on cancer care disparities and offers actionable data for improving healthcare policy and practices in similar contexts.

# Conclusion and recommendations

## Conclusions

The study's valuable information lies in identifying the key factors contributing to non-adherence among cervical cancer patients: inadequate healthcare infrastructure, financial constraints, malfunctioning radiotherapy equipment, and insufficient patient education. This information is valuable as it highlights specific barriers that prevent effective cancer care and lead patients to seek self-medication including traditional remedies. The importance of addressing these barriers cannot be overstated—improving healthcare access, funding, equipment reliability, and patient education are essential steps to enhance treatment adherence, reduce the burden on cancer services, and improve patient outcomes. These insights provide actionable targets for health policy interventions aimed at improving cervical cancer care in resource-limited settings. Cervical cancer patients in the Lake and Western zones who seek treatment at BMC often turn to nearby healthcare facilities, which, unfortunately, have limited capacity as far as cancer management is concerned. As a result, some resort to traditional remedies or self-medications. There is a pressing need for improved access to appropriate healthcare for cervical cancer patients in the study area. Financial constraints pose a substantial challenge to treatment adherence. To surmount these obstacles, essential steps must be taken to help cervical cancer patients, for example, improving funding options such as including cervical cancer services in the improved community health fund and the forthcoming universal health coverage to cater for their treatment and other related costs.

Comprehensive information, education, and communication are insufficient (IEC) on the part of patients regarding cervical cancer which exacerbates difficulties in adhering to treatment plans. IEC efforts in disseminating pertinent information to patients should be intensified. Limitations imposed by a malfunctioning radiotherapy machine are immense and costly on the part of patients. Rectifying the radiotherapy machine's malfunction in time and the possibility of acquiring another machine as a backup should implemented. The above-mentioned measures are pivotal in not only addressing non-adherence but also elevating the overall standard of care and augmenting outcomes for cervical cancer patients in the area.

## Study strengths

Utilizing a phenomenological design allowed us to undertake in-depth exploration which provided a very rich, detailed description and deeper understanding of the phenomenon as experienced by informants (lived experiences of cervical cancer patients), making it particularly

well-suited for comprehending the subjective aspects of their healthcare-seeking behaviors and adherence to treatment by within the community where the patients reside ensures that the findings are grounded in the local context, potentially leading to more applicable and actionable recommendations. By centering on the experiences of cervical cancer patients, the study prioritizes the perspectives of those directly affected, which is crucial for designing effective interventions.

## Study limitations

The purposive sampling of phenomenological design used in this study is usual practice in qualitative research because the findings are inherently not generalizable. A sample of 15 non-adherent cervical cancer patients is not large enough to make definitive generalizations that go beyond the research site.

Moreover, the accuracy and completeness of medical records affected the study's ability to obtain detailed information about patients' treatment and adherence history, particularly the absence of a clear chemotherapy treatment plan indicating how many sessions the patient was supposed to receive and when. Despite these limitations, our study still has demonstrated the value of using a qualitative study designed to explore the lived experience of cervical cancer patients in the study areas.

## Recommendations

Based on the findings of the study, we recommend the following:

The Ministry of Health's Directorate of Diagnostic Services should establish and enhance diagnostic capabilities for cervical cancer within lower-ranking healthcare facilities. This should include ensuring the availability of skilled personnel for accurate and timely cancer diagnoses and facilitating the prompt referral of patients to specialized cancer care hospitals for proper management. The Ministry of Health's Directorate of Medical Services should institute financial aid and subsidies to ease treatment costs, prioritizing vulnerable groups. This encompasses health coverage for low-income individuals, including cancer treatments, and charity care programs for those unable to cover medical expenses and related treatment costs.

BMC Oncology Department should ensure the constant availability of cancer treatments by conducting consistent and rigorous maintenance of critical medical equipment to prevent malfunctions and disruptions in treatment schedules. Backup essential machines should be available. Cervical cancer stakeholders such as NGOs and Public Health Specialists should conduct targeted and well-coordinated IEC activities. Health education programs through activity-based methods such as drama, concerts, etc., electronic, print media, and social media, raise awareness about cervical cancer, emphasizing the importance of seeking professional medical care over traditional remedies and self-medication.

Bugando Oncology Department should establish a cancer patient support and information unit that provides clear, accessible information regarding treatment schedules and required interventions, alongside offering counseling on treatment expectations, potential side effects, and adherence to guidelines.

## Recommendations for further studies

Future studies should involve a broader range of stakeholders, including healthcare providers, caregivers, and family members, to provide a more holistic perspective on healthcare-seeking behaviors and adherence patterns among cervical cancer patients. Intervention studies targeting specific aspects of healthcare-seeking behavior and adherence could yield valuable insights into effective strategies for improving patient compliance with treatment plans.

## Supporting information

**S1 File. Consent form.**
(PDF)

**S2 File. Raw data.**
(PDF)

**S3 File. Code book.**
(DOCX)

**S4 File. Results summary.**
(XLSX)

## Acknowledgments

We acknowledge the School of Public Health at the Catholic University of Health and Allied Sciences, and Bugando Medical Centre, for their collaboration and assistance with this study. We thank the Regional Administrative and Local Government Authorities in Mwanza, Tanzania for their practical support. We express our sincere gratitude to all the non-adherent cervical cancer patients for their participation in this study.

## Author contributions

**Conceptualization:** Joseph Rogathe Mwanga, Bashari Nuru Kidaya, Halima Mdemu Mwaisungu.

**Data curation:** Bashari Nuru Kidaya, Elisha Mabula Juma, Vivian Elikana Buremo, Halima Mdemu Mwaisungu.

**Formal analysis:** Joseph Rogathe Mwanga, Bashari Nuru Kidaya, Oscar Ottoman Muhini, Elisha Mabula Juma, Halima Mdemu Mwaisungu.

**Funding acquisition:** Bashari Nuru Kidaya.

**Investigation:** Joseph Rogathe Mwanga, Bashari Nuru Kidaya, Vivian Elikana Buremo.

**Methodology:** Joseph Rogathe Mwanga, Bashari Nuru Kidaya, Oscar Ottoman Muhini.

**Project administration:** Joseph Rogathe Mwanga.

**Resources:** Bashari Nuru Kidaya.

**Supervision:** Joseph Rogathe Mwanga, Oscar Ottoman Muhini, Halima Mdemu Mwaisungu.

**Writing – original draft:** Joseph Rogathe Mwanga, Bashari Nuru Kidaya, Oscar Ottoman Muhini.

**Writing – review & editing:** Joseph Rogathe Mwanga, Oscar Ottoman Muhini, Elisha Mabula Juma, Vivian Elikana Buremo, Halima Mdemu Mwaisungu.

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
