## [Decision Letter · Decision Letter 0]

23 Sep 2024

Dear Dr. Mwanga,

Thank you for submitting your manuscript to PLOS ONE. After careful consideration, we feel that it has merit but does not fully meet PLOS ONE’s publication criteria as it currently stands. Therefore, we invite you to submit a revised version of the manuscript that addresses the points raised during the review process.

The manuscript does not meet the standards of academic writing. Please ensure that you include the steps taken to ensure trustworthiness in the manuscript. The discussion requires revision due to its excessive length and redundant content. Update the reference, especially in relation to disease statistics. 

We look forward to receiving your revised manuscript.

Kind regards,

Roaa Sabri Gassas

Academic Editor

PLOS ONE

2. We note that your Data Availability Statement is currently as follows: [All relevant data are within the manuscript and its Supporting Information files.] Please confirm at this time whether or not your submission contains all raw data required to replicate the results of your study. Authors must share the “minimal data set” for their submission. PLOS defines the minimal data set to consist of the data required to replicate all study findings reported in the article, as well as related metadata and methods (https://journals.plos.org/plosone/s/data-availability#loc-minimal-data-set-definition). For example, authors should submit the following data: - The values behind the means, standard deviations and other measures reported; - The values used to build graphs; - The points extracted from images for analysis. Authors do not need to submit their entire data set if only a portion of the data was used in the reported study. If your submission does not contain these data, please either upload them as Supporting Information files or deposit them to a stable, public repository and provide us with the relevant URLs, DOIs, or accession numbers. For a list of recommended repositories, please see https://journals.plos.org/plosone/s/recommended-repositories. If there are ethical or legal restrictions on sharing a de-identified data set, please explain them in detail (e.g., data contain potentially sensitive information, data are owned by a third-party organization, etc.) and who has imposed them (e.g., an ethics committee). Please also provide contact information for a data access committee, ethics committee, or other institutional body to which data requests may be sent. If data are owned by a third party, please indicate how others may request data access.

Additional Editor Comments (if provided):

Reviewers' comments:

Reviewer's Responses to Questions

**Comments to the Author**

1. Is the manuscript technically sound, and do the data support the conclusions?

Reviewer #1: Yes

Reviewer #2: Yes

2. Has the statistical analysis been performed appropriately and rigorously?

Reviewer #1: Yes

Reviewer #2: Yes

3. Have the authors made all data underlying the findings in their manuscript fully available?

Reviewer #1: Yes

Reviewer #2: Yes

4. Is the manuscript presented in an intelligible fashion and written in standard English?

Reviewer #1: No

Reviewer #2: Yes

Reviewer #1: This manuscript addresses a relevant topic, emphasizing the healthcare-seeking behaviors and factors influencing non-adherence among cervical cancer patients attending bugando oncology clinic in Mwanza, Tanzania. The manuscript is well conducted and evidenced by data. However, the manuscript contains some grammar errors and awkward phrasing. A comprehensive language review could improve readability.

Reviewer #2: Comments to the Author

Congratulations on the submitted manuscript. The topic is timely and will be of interest to the readers of the journal. However, a few changes are suggested to improve the clarity of this manuscript. I have several recommendations and questions about the manuscript.

Need to paraphrase because the similarity index is 29 using Turnitin

Abstract

Findings:

The findings revealed delayed healthcare-seeking behavior and poor adherence to most of the study participants seeking medical care at nearby health facilities with limited capacity for correct cervical cancer diagnosis. Some turned to self

medications including traditional remedies.

-Please harmonize the statements.

Introduction

Among the estimated 40,464 individuals diagnosed with cancer in 2020, cervical cancer constituted 25.3% of all newly identified cases.

-Please add a reference to support this statement.

Methodology

In this research we took into consideration the process known as ‘bracketing’

-Explain the types of Bracketing used here, Descriptive Bracketing or Reflexive Bracketing?

Kiswahili transcripts were then translated into English for further processing.

-Explain in detail the further processing after translation, the English transcripts regarding thematic analysis, coding and report writing in subheading, to make clearer.

Results

These individuals included women aged 35–39 (n=2), 40–49 (n=3),

50–59 (n=8), and 60–65 (n=2). Their education levels ranged from no formal education (n=5) to

primary school dropout (n=3), primary school completion (n=6), and secondary school dropout

(n=1). Marital statuses included single (n=1), married (n=6), separated (n=1), divorced (n=3), and

widowed (n=4). Religious affiliations varied, with informants identifying as having no religion

(n=2), Catholic (n=6), Mennonite (n=1), Pentecostal (n=3), Baptist (n=1), and Muslim (n=2).

-Please write in percentages also example: n(%).

Discussion

- Highlight the implication of the results to health policy

- How do the findings add to the body of scientific knowledge on the issue?

Conclusion

-The conclusion very general.

-The author should specify what was the 'valuable information' in the conclusion and briefly describe the importance of the 'valuable information".

References

-16/ 34 of the references 5 year above.

-Recommended for 5 years and under

**Do you want your identity to be public for this peer review?** For information about this choice, including consent withdrawal, please see our Privacy Policy

Reviewer #1: **Yes: ** DR. YAHYA H. Y. ALFARRA, BDS (Hons), MSc, PhD

Reviewer #2: **Yes: ** DR RUSNANI AB LATIF

---

## [Author Response · Author response to Decision Letter 0]

20 Nov 2024

We have responded to the Reviewers' comments as per attached Rebuttal letter.

---

## [Decision Letter · Decision Letter 1]

2 Jan 2025

‘Healthcare-seeking behaviors and factors influencing non-adherence among cervical cancer patients attending Bugando oncology clinic in Mwanza, Tanzania: A qualitative phenomenological study.’

PONE-D-24-22844R1

Dear Dr. Mwanga,

We’re pleased to inform you that your manuscript has been judged scientifically suitable for publication and will be formally accepted for publication once it meets all outstanding technical requirements.

Kind regards,

Roaa Sabri Gassas

Academic Editor

PLOS ONE

Additional Editor Comments (optional):

Reviewers' comments:

Reviewer's Responses to Questions

**Comments to the Author**

Reviewer #1: All comments have been addressed

2. Is the manuscript technically sound, and do the data support the conclusions?

Reviewer #1: Yes

3. Has the statistical analysis been performed appropriately and rigorously?

Reviewer #1: Yes

4. Have the authors made all data underlying the findings in their manuscript fully available?

Reviewer #1: Yes

5. Is the manuscript presented in an intelligible fashion and written in standard English?

Reviewer #1: Yes

Reviewer #1: The authors amended the original manuscript to incorporate the recommendations of the reviewers. Therefore, I have no further suggestions for this article.

**Do you want your identity to be public for this peer review?** For information about this choice, including consent withdrawal, please see our Privacy Policy

Reviewer #1: **Yes: ** DR. YAHYA H. Y. ALFARRA, BDS (Hons), MSc, PhD

---

## [Editor Report · Acceptance letter]

PONE-D-24-22844R1

PLOS ONE

Dear Dr. Mwanga,

I'm pleased to inform you that your manuscript has been deemed suitable for publication in PLOS ONE. Congratulations! Your manuscript is now being handed over to our production team.

Kind regards,

on behalf of

Dr. Roaa Sabri Gassas

Academic Editor

PLOS ONE